# Trends and cross-country inequities by region, sex, age in the mortality, incidence, and disability-adjusted life years of COVID-19: Analysis from the Global Burden of Disease Study 2021

**Xiao-Han Xu[1,2], Jing Sun[3,4,5,6], Jin-Lun Zhang[1], Zhou Yang[1]\*, Chun-Quan Ou[1]\***

**1** State Key Laboratory of Multi-organ Injury Prevention and Treatment, Department of Biostatistics, School of Public Health, Guangdong Provincial Key Laboratory of Tropical Disease Research, Southern Medical University, Guangzhou, China, **2** Department of Epidemiology and Health Statistics, School of Public Health, Guangdong Medical University, Dongguan, China, **3** Rural Health Research Institute, Charles Sturt University, Orange, New South Wales, Australia, **4** Honorary Professor, Data Science Institute, University of Technology Sydney, Sydney, New South Wales, Australia, **5** Honorary Professor, Heart Research Institute, University of Sydney, Sydney, New South Wales, Australia, **6** Adjunct professor, School of Health Sciences and Social Work, Griffith University, Brisbane, Queensland, Australia

\* ouchunquan@hotmail.com (CQO); yang_zhoucn@163.com (ZY)

## Abstract

### Background

The coronavirus disease 2019 (COVID-19) pandemic has imposed a substantial disease burden globally and has further exacerbated pre-existing health inequities. This study aimed to provide a comprehensive assessment of the burden and inequities associated with COVID-19 across diverse populations.

### Methods

Using data from the Global Burden of Disease 2021, we systematically analyzed deaths, incidence, disability-adjusted life years (DALYs), and years of life lost (YLLs) of COVID-19 stratified by sex, age, and region. The temporal trends pre- and post-2019 (i.e., 1990–2019 and 2019–2021) were measured using average annual percent change (AAPC). Additionally, the cross-country absolute and relative socio-demographic index (SDI)-related health inequities were assessed using the slope index and concentration index, respectively. The SDI is a composite development indicator that incorporates income, educational attainment, and fertility conditions.

### Results

Trends in the global burden of all-cause mortality and DALYs exhibited significant declines (AAPC < 0) from 1990 to 2019 but underwent a marked reversal trend

**Data availability statement:** Data used for the analyses can be accessed openly through the GBD 2021 online database (https://vizhub.healthdata.org/gbd-results/).

**Funding:** This work was supported by the National Natural Science Foundation of China (grant numbers: 82373679 and 81973140 to CQO; 82504514 to ZY; https://www.nsfc.gov.cn/). The funders played no role in study design, data collection and analysis, decision to publish, or preparation of the manuscript.

**Competing interests:** The authors have declared that no competing interests exist.

(AAPC > 0) following the COVID-19 pandemic. In 2021, COVID-19 resulted in 2.28 billion incident cases and 7.89 million deaths globally, with an age-standardized DALYs rate of 2,501 per 100,000 population. While incidence rates were relatively evenly distributed across populations, mortality was disproportionately higher among males and older adults. Substantial health inequities in the burden of COVID-19 were evident across 204 countries and territories, with absolute widening inequities notably in 2021 (e.g., the slope index of inequity for DALYs rose from 2,713 in 2020 to 4,044 in 2021). Greater inequities are disproportionately concentrated among males, middle-aged and older individuals, and regions with lower SDI levels.

## Conclusions

These findings highlight the substantial disease burden of COVID-19 and elucidate the multidimensional health inequities exacerbated by the pandemic, providing crucial evidence for targeted interventions to address inequities and strengthen resilience in future global health emergencies.

## Author summary

The coronavirus disease 2019 (COVID-19) pandemic has imposed a substantial disease burden globally in recent years. Evaluating health inequities associated with COVID-19 is essential for strengthening the current healthcare system and guiding public health policy priorities to inform future global health emergencies. We found that the all-cause disease burden had significantly declined prior to the COVID-19 pandemic but subsequently reversed due to the substantial impact of COVID-19, resulting in an upward trend from 2019 to 2021. Although the incidence of COVID-19 was relatively consistent across sex and age groups, mortality and disability-adjusted life years were markedly higher among males and older people. Substantial absolute and relative inequities associated with the sociodemographic index were observed in the burden of COVID-19 across 204 countries and territories, with these inequities further widening in 2021. Furthermore, greater health inequities were disproportionately observed among males, middle-aged and older populations, and regions with a lower sociodemographic index. These findings underscore the profound global impact of COVID-19 and its associated health inequities across multiple dimensions, highlighting the critical need for targeted public health interventions to mitigate inequities and strengthen the resilience of health systems in anticipation of future global health emergencies.

## Introduction

The coronavirus disease 2019 (COVID-19) pandemic has exerted profound and multi-dimensional impacts on global health, economies, and societies, straining

public health systems, necessitating unprecedented resources mobilization and policy adaptations to curb the rapid spread of the virus [1]. This pandemic may have the potential to highlight and exacerbate existing health inequities across social, economic, and demographic groups [2,3], thereby reflecting the unequal distribution of health outcomes across diverse populations [4]. Consequently, investigating the interplay of COVID-19 incidence and mortality with pre-existing health inequities has become a critical research priority. Such research can illuminate how effectively healthcare systems manage global health crises from an equity perspective and support the development of more inclusive health systems to confront future emergencies effectively [3,5].

Historically, health inequities have intensified the adverse health outcomes of marginalized populations attributable to pandemics [6]. For example, during the Black Death, impoverished communities experienced disproportionately high mortality rates, primarily attributable to insufficient sanitation, overcrowded living conditions, and limited access to medical care [7]. Similarly, during the 1918 Spanish flu pandemic, urban slum areas experienced disproportionately high infection and mortality rates [8]. Despite recent advances in global economic conditions and healthcare systems, such inequities have persisted across subsequent pandemics, including the H1N1 influenza pandemic in England [9], the Zika syndrome outbreak in Brazil [10], and Ebola outbreaks in West Africa [11]. The COVID-19 pandemic has starkly exposed persistent health inequities. A growing body of evidence demonstrates that socioeconomic status, race and ethnicity, age, and sex are key determinants influencing the incidence, severity, and mortality of COVID-19 [3,12,13]. In the United States, racial and ethnic minorities, particularly African Americans and Hispanics, experienced disproportionately higher COVID-19 infection and mortality rates compared to White populations, driven by chronic health conditions, limited healthcare access, and other factors [14]. Similar patterns were observed in Europe, where individuals in lower-income groups faced increased vulnerability due to overcrowded living conditions and limited access to protective resources [3].

Although considerable research has examined the health impacts of COVID-19, most studies to date have focused on specific nations or regions, predominantly high-income countries, and lack the global perspective to elucidate cross-country differences and patterns of inequity [4,6]. Furthermore, limited evidence exists on how age and sex intersect with health inequities to affect COVID-19 outcomes globally, especially in low- and middle-income countries. To address these gaps, this study investigated cross-country inequities in COVID-19 incidence, mortality, and burden, utilizing data from the Global Burden of Disease Study 2021 (GBD 2021). The GBD 2021 provided age- and sex-specific burden of COVID-19 across 204 countries and territories during the first two years of the COVID-19 pandemic (2020–2021) [13]. Previous studies based on GBD 2021 have mainly focused on general descriptive analyses of the COVID-19 disease burden [13,15] or the influence of the COVID-19 pandemic on other diseases, such as malaria [16], stroke [17], and depressive disorders [18]. However, quantitative evidence regarding the socioeconomic inequities in the burden of COVID-19 remains scarce. Leveraging the comprehensive scope of GBD 2021 data, this systematic analysis offers a unique perspective on variations across multiple dimensions, providing a nuanced understanding of how pre-existing health inequities have led to disparate COVID-19 outcomes [13]. Thus, addressing these inequities is vital not only to mitigate the current pandemic's impact but also to enhance resilience against future global health emergencies.

## Methods

### Overview

The GBD 2021 provided a comprehensive assessment of age- and sex-specific health loss for 371 diseases and injuries, 288 causes of death, and 88 risk factors in 204 countries and territories from 1990 to 2021 based on updated epidemiological data and improved estimation methodologies [13,15]. Notably, the latest version of the GBD had important updates, such as a new input data source for COVID-19. The GBD 2021 introduced two causes related to COVID-19: COVID-19 (Level 3), which denotes the direct effects of the disease, including long-term COVID as a sequela of infection,

and other pandemic-related mortality (OPRM, Level 1), which refers to excess mortality that cannot be attributed to specific causes (i.e., COVID-19 and four other pandemic-related causes) [13]. Our study primarily focused on the disease burden associated with COVID-19.

## Data input and modeling

The GBD framework integrates multiple data sources and applies standardized ensemble modeling strategies to estimate the global burden of COVID-19 [13,19]. The detailed description of the disease burden assessment methodology for COVID-19 has been published previously [13,19].

Briefly, the estimation of COVID-19 incidence leveraged diverse sources of data inputs, including reported cases, hospital admissions, and reported deaths (primarily from national databases and the Johns Hopkins University COVID-19 dashboard), as well as representative seroprevalence surveys identified through systematic reviews, SeroTracker, and governmental platforms. These data were adjusted for known biases, such as reporting delays, underreporting of deaths, waning antibody sensitivity, vaccination coverage, and reinfections. Infection-detection ratios, infection-hospitalization ratios, and infection-fatality ratios were empirically derived. For locations lacking input data, predictive models incorporating published covariates were developed to estimate daily infections.

The estimation of COVID-19-related mortality was based on weekly or monthly all-cause mortality data from vital registration and surveillance reports. The International Classification of Diseases, 10th revision (ICD-10) codes for COVID-19 were defined as U07–U07.2. To address incomplete mortality reporting across regions, GBD employed a two-stage modeling framework that enables robust and comparable estimates across countries. For regions with input data, excess mortality was calculated as the difference between observed and expected mortality, the latter estimated using counterfactual statistical models assuming the pandemic did not occur. For regions without sufficient input data, a Bayesian model averaging (Rover model) was used to predict excess mortality based on 11 covariates, including COVID-19 pandemic covariates (e.g., seroprevalence) and background population health metrics (e.g., Healthcare Access and Quality Index). Excess deaths were then calculated by multiplying excess mortality rates by the person-year exposure for the period.

We collected the number of cases and deaths, along with their rates (per 100,000 population) and corresponding 95% confidence intervals (CIs) of COVID-19 by sex, age, region, and country from 2020 to 2021. According to the GBD 2021 algorithm, these intervals were calculated as the 2.5th and 97.5th percentiles of 1,000 or 500 draw-level estimates for each parameter generated from posterior or simulation-based distributions [13,15,20]. Age-standardized rates (ASRs) were determined using the age structure derived from the GBD standard population to obtain a weighted average for cross-country comparisons. In addition, the GBD employed multiple metrics to quantify the impact of COVID-19, including incidence, deaths, disability-adjusted life years (DALYs), and years of life lost (YLLs). YLLs were derived by multiplying the number of deaths by the remaining life expectancy at the standardized age of death according to the life table in GBD 2021. The composite measure of DALYs was calculated as the sum of YLLs and years lived with disability (YLDs), which include both acute and long-term disability resulting from COVID-19. Regional estimates were constructed by aggregating countries based on geographical proximity and epidemiological similarity, comprising 7 super-regions, 21 regions, and 204 countries and territories. The sociodemographic index (SDI) is a composite measure developed by the GBD collaborators to reflect the socioeconomic and demographic development of a country or region [13]. It is calculated as the geometric mean of three rescaled indicators: lag-distributed income per capita, mean educational attainment among individuals aged 15 and older, and total fertility rate among women under the age of 25. Each component is normalized to a scale from 0 to 1 using empirically derived bounds. The 204 countries and territories were then categorized into five regions based on the SDI: low, low-middle, middle, high-middle, and high. Moreover, we collected all-cause deaths and DALYs to explore changes in temporal trends pre- and post-2019 (i.e., 1990–2019 and 2019–2021). The data were publicly accessible

through web-based tools (available from: https://vizhub.healthdata.org/gbd-results/) provided by the Institute for Health Metrics and Evaluation (IHME) and the Global Health Data Exchange (GHDx). All data used in this study were extracted on July 13, 2024. This study followed the guidelines for accurate and transparent health estimates reporting (GATHER) and GBD study guidelines.

### Trend analysis

Joinpoint regression analysis was employed to examine temporal trends in the all-cause burden pre- and post-2019 (i.e., 1990–2019 and 2019–2021) (https://surveillance.cancer.gov/joinpoint/). Although the first COVID-19 case was identified in December 2019, the pandemic began in 2020. The year 2019 was chosen as the predefined breakpoint because it marks the end of the pre-pandemic trend and allows for direct comparison between the subsequent years with 2019 to assess substantial changes in disease burden during the COVID-19 pandemic (2020–2021). The grid search method was applied to identify turning points (i.e., joinpoints) in the trends. Then, the average annual percentage change (AAPC) and 95% CI were calculated by weighted averaging of the annual percent changes across subintervals. We also calculated the percentage change in the burden of COVID-19 between 2020 and 2021. An AAPC > 0 indicates an increasing trend over a given period, while an AAPC < 0 indicates a decreasing trend. The burden is considered stable if the CI contains zero.

### Cross-country inequities analysis

The slope index and concentration index are widely used to assess absolute and relative health inequity within the GBD framework. We used both indexes to quantify SDI-related inequity in the COVID-19 burden across countries. The slope index of inequity was determined by linearly regressing country-level ASRs of COVID-19 against the cumulative population proportion after sorting all countries by SDI [21]. Then, the absolute difference between the predicted ASR values at the proportions of 1 and 0 represents the slope index. The concentration index was calculated as twice the area between the equality line (i.e., the 45-degree diagonal) and the Lorenz concentration curve, which was fitted to the observed cumulative relative distributions of the SDI-ranked population and ASR. The concentration index ranges from −1 to +1, with absolute values closer to 0 indicating greater equity. A negative concentration index (i.e., the concentration curve above the equity line) suggests that the disease burden is concentrated in regions and countries with lower SDI. Conversely, a positive value (i.e., the concentration curve below the equity line) indicates a higher accumulation of disease burden in higher SDI regions and countries. Since our study focused on health inequities due to SDI differences, the ASR was used to minimize the confounding effect of the age structure of populations. Moreover, we further explored health inequities associated with the three components of the SDI for a more detailed assessment.

### Results

Table 1 and S1 Fig show the temporal trends for all-cause disease burden before and after 2019 (i.e., 1990–2019 and 2019–2021). The age-standardized mortality and DALY rates presented a significant downward trend from 1990 to 2019 (AAPC < 0), while both increased substantially during the pandemic period (AAPC > 0). The reversal was particularly striking for age-standardized mortality rates, with the AAPC changing from −1.39 (95% CI: −1.47 to −1.30) to 6.82 (95% CI: 5.49 to 8.18). The change in AAPC for both mortality and DALYs was greater in males than in females. Before COVID-19, the AAPC in mortality rates was relatively balanced across the five SDI-based regions, whereas the AAPC increased with decreasing SDI during 2019–2021. For DALYs, regions with lower SDI levels demonstrated a greater annual decrease before COVID-19 but experienced a more rapid increase during the pandemic period. Notably, after excluding the cases and deaths of COVID-19, the AAPCs during the pandemic period were not statistically significant.

**Table 1. Age-standardized rates and average annual percentage change of death and DALYs for all-cause by sex and SDI between 1990 and 2021.**

| | Age-standardized rate per 100,000 (95% CI) | | | Average annual percentage change (95% CI) | | |
|---|---|---|---|---|---|---|
| | 1990 | 2019 | 2021 | 1990–2019 | 2019–2021 | 2019–2021 (COVID-19 exclusion) |
| **Death** | | | | | | |
| Global | 1,107(1,083–1,133) | 735(707–768) | 835(801–870) | −1.39(−1.47 to −1.30) | 6.82(5.49 to 8.18) | 0.41 (−2.07 to 2.95) |
| Sex | | | | | | |
| Male | 1,296(1,259–1,340) | 878(835–923) | 1,019(970–1,071) | −1.33(−1.43 to −1.23) | 8.18(7.17 to 9.19) | 0.83 (−0.78 to 2.47) |
| Female | 952(927–979) | 614(585–645) | 681(649–714) | −1.49(−1.59 to −1.39) | 5.34(4.49 to 6.20) | −0.07 (−3.64 to 3.63) |
| SDI | | | | | | |
| Low SDI | 1,819(1,745–1,893) | 1,181(1,098–1,276) | 1,478(1,375–1,582) | −1.43(−1.60 to −1.26) | 12.11(10.19 to 14.06) | 2.30 (−3.47 to 8.42) |
| Low-middle SDI | 1,378(1,339–1,417) | 959(914–1,005) | 1,151(1,095–1,207) | −1.22(−1.48 to −0.96) | 9.78(7.08 to 12.55) | 1.21 (−4.12 to 6.84) |
| Middle SDI | 1,106(1,061–1,154) | 719(682–762) | 819(773–867) | −1.49(−1.68 to −1.30) | 6.95(5.86 to 8.05) | 0.64 (−4.55 to 6.10) |
| High-middle SDI | 973(941–1,006) | 621(587–657) | 690(651–729) | −1.50(−1.76 to −1.25) | 5.70(3.09 to 8.37) | −0.19 (−5.79 to 5.75) |
| High SDI | 693(690–697) | 440(436–445) | 479(474–484) | −1.52(−1.59 to −1.45) | 4.40(3.11 to 5.70) | −0.82 (−3.57 to 2.01) |
| **DALY** | | | | | | |
| Global | 50,766(47,546–54,275) | 33,760(30,670–37,223) | 36,203(33,062–39,614) | −1.39(−1.45 to −1.32) | 3.65(2.75 to 4.56) | −0.08 (−1.72 to 1.58) |
| Sex | | | | | | |
| Male | 55,203(52,168–58,298) | 36,688(33,693–40,155) | 40,044(36,910–43,464) | −1.40(−1.47 to −1.33) | 4.84(4.17 to 5.52) | 0.15 (−1.26 to 1.58) |
| Female | 46,749(43,312–50,816) | 31,070(27,787–34,865) | 32,691(29,265–36,270) | −1.40(−1.49 to −1.30) | 2.63(1.88 to 3.39) | −0.32 (−2.24 to 1.62) |
| SDI | | | | | | |
| Low SDI | 87,820(83,112–92,538) | 51,947(47,341–57,809) | 57,892(52,792–63,937) | −1.76(−1.89 to −1.62) | 5.75(3.46 to 8.09) | 0.29 (−3.41 to 4.13) |
| Low-middle SDI | 64,006(60,440–67,832) | 40,778(37,357–44,779) | 44,819(41,446–48,966) | −1.53(−1.63 to −1.44) | 4.85(3.96 to 5.76) | −0.10 (−3.39 to 3.3) |
| Middle SDI | 45,738(42,631–49,335) | 29,420(26,520–33,009) | 31,792(28,845–35,014) | −1.52(−1.60 to −1.44) | 4.27(3.67 to 4.87) | 0.07 (−2.79 to 3.02) |
| High-middle SDI | 37,438(34,492–40,585) | 24,109(21,259–27,186) | 25,775(23,139–28,837) | −1.50(−1.66 to −1.33) | 3.73(2.05 to 5.43) | −0.09 (−1.88 to 1.73) |
| High SDI | 27,976(25,240–31,147) | 21,326(18,604–24,584) | 22,603(19,871–25,892) | −0.92(−0.97 to −0.87) | 3.00(2.15 to 3.86) | 0.13 (−2.28 to 2.61) |

DALY, Disability-Adjusted Life Year; SDI, sociodemographic index; CI, confidence interval.

In 2021, the global age-standardized incidence rate of COVID-19 was 28,955 (95% CI: 27,708–30,143) per 100,000 population, with a total of 2.28 (95% CI: 2.18–2.37) billion cases and 7.89 (95% CI: 7.51–8.40) million deaths worldwide (Table 2). COVID-19 resulted in 212.01 (95% CI: 197.94–234.67) million DALYs and 197.76 (95% CI: 187.90–211.51) million YLLs, respectively (S1 Table). YLLs accounted for 93.27% of total DALYs, indicating that premature death was the predominant component of the disease burden attributable to COVID-19. The ASRs of the four indicators (i.e., incidence, deaths, DALYs, and YLLs) consistently increased with decreasing SDI, such as incidence (low SDI region: 44,162 per 100,000 versus high SDI region: 14,188 per 100,000) and DALYs (low SDI region: 5,646 per 100,000 versus high SDI region: 1,220 per 100,000).

In 2021, the global age-standardized incidence rate of COVID-19 was 28,699 (95% CI: 27,461–29,887) per 100,000 population for males and 29,213 (95% CI: 27,954–30,402) per 100,000 population for females (Table 2). Except for the under-5 age group, which showed a lower number and rate of COVID-19, the number of incident cases gradually decreased with increasing age (Fig 1). The incidence rates were similar across age groups between 5 and 64 years, with a relatively high rate in the 60–64 age group, and then decreased as age further increased. Moreover, the number and rates of incidence were almost equal in both sexes. However, the DALYs demonstrated marked sex-specific differences, with males (3,248 per 100,000 population, 95% CI: 3,060–3,519) having a higher burden than females (1,823 per 100,000 population, 95% CI: 1,651–2,127), and this disparity persisted consistently across all age groups. In addition, the 65–69 age group had the largest number of DALYs (23.84 million, 95% CI: 22.63–25.53) due to COVID-19 in 2021, whereas

**Table 2. Number of cases and age-standardized rates of incidence, death, and DALYs for COVID-19 by sex and SDI in 2020 and 2021.**

| | 2020 | | 2021 | | Percentage |
|---|---|---|---|---|---|
| | Number (95% CI, *10⁶) | Age-standardized rate per 100,000 (95% CI) | Number (95% CI, *10⁶) | Age-standardized rate per 100,000 (95% CI) | Change in age-standardized rates between 2020 and 2021 (95% CI) |
| **Incidence** | | | | | |
| Global | 1,625.53(1,554.21–1,693.21) | 20,820(19,908–21,681) | 2,279.72(2,181.36–2,372.46) | 28,955(27,708–30,143) | 39.08(31.00 to 47.65) |
| Sex | | | | | |
| Male | 813.28(777.44–847.19) | 20,663(19,754–21,524) | 1,139.12(1,089.86–1,186.08) | 28,699(27,461–29,887) | 38.89(30.78 to 47.51) |
| Female | 812.25(776.82–845.84) | 20,982(20,067–21,849) | 1,140.59(1,091.70–1,186.38) | 29,213(27,954–30,402) | 39.23(31.16 to 47.79) |
| SDI | | | | | |
| Low SDI | 342.88(315.55–367.20) | 32,146(29,567–34,502) | 480.82(436.28–517.97) | 44,162(39,935–47,666) | 37.38(22.28 to 54.34) |
| Low-middle SDI | 583.68(524.33–635.83) | 30,419(27,315–33,130) | 783.32(707.00–862.84) | 40,191(36,266–44,240) | 32.13(15.12 to 51.64) |
| Middle SDI | 419.27(397.96–440.59) | 17,224(16,342–18,099) | 593.57(562.04–624.05) | 24,294(23,014–25,540) | 41.05(31.14 to 51.71) |
| High-middle SDI | 160.43(145.37–179.11) | 12,208(11,038–13,639) | 268.35(247.81–289.25) | 20,549(19,029–22,158) | 68.31(47.65 to 91.87) |
| High SDI | 118.21(112.77–124.73) | 11,177(10,658–11,801) | 151.60(145.96–157.61) | 14,188(13,649–14,767) | 26.93(19.00 to 35.39) |
| **Death** | | | | | |
| Global | 4.80(4.57–5.11) | 59(56–62) | 7.89(7.51–8.40) | 94(89–100) | 60.16(47.98 to 73.34) |
| Sex | | | | | |
| Male | 2.97(2.81–3.17) | 80(76–86) | 4.80(4.55–5.14) | 126(120–135) | 57.19(44.64 to 70.82) |
| Female | 1.83(1.74–1.95) | 41(39–43) | 3.08(2.92–3.28) | 67(63–71) | 64.39(51.61 to 78.26) |
| SDI | | | | | |
| Low SDI | 0.63(0.60–0.68) | 146(138–157) | 1.06(1.00–1.13) | 242(227–256) | 65.85(51.98 to 80.98) |
| Low-middle SDI | 1.40(1.31–1.49) | 110(103–117) | 2.20(2.07–2.37) | 169(159–182) | 53.08(39.53 to 67.95) |
| Middle SDI | 1.26(1.16–1.39) | 52(48–58) | 2.26(2.09–2.51) | 91(84–101) | 74.17(52.9 to 98.41) |
| High-middle SDI | 0.71(0.65–0.77) | 38(35–41) | 1.38(1.27–1.49) | 72(66–77) | 88.96(68.49 to 111.92) |
| High SDI | 0.80(0.77–0.82) | 38(37–39) | 0.98(0.95–1.01) | 46(44–48) | 20.56(14.75 to 26.67) |
| **DALY** | | | | | |
| Global | 123.35(116.49–132.67) | 1,482(1,399–1,595) | 212.01(197.94–234.67) | 2,501(2,330–2,778) | 68.73(50.97 to 88.58) |
| Sex | | | | | |
| Male | 78.53(74.21–84.28) | 1,979(1,870–2,123) | 131.77(124.12–142.71) | 3,248(3,060–3,519) | 64.13(49.23 to 80.52) |
| Female | 44.82(41.78–49.37) | 1,033(960–1,145) | 80.24(73.12–92.52) | 1,823(1,651–2,127) | 76.41(50.56 to 106.70) |
| SDI | | | | | |
| Low SDI | 19.27(17.88–21.04) | 3,347(3,133–3,626) | 33.65(30.87–38.22) | 5,646(5,246–6,168) | 68.72(51.14 to 88.34) |
| Low-middle SDI | 38.82(36.02–42.31) | 2,602(2,417–2,807) | 63.42(58.15–71.19) | 4,123(3,820–4,559) | 58.48(41.00 to 78.13) |
| Middle SDI | 33.37(30.70–36.99) | 1,280(1,178–1,418) | 62.10(56.80–70.33) | 2,329(2,131–2,638) | 81.96(57.60 to 110.09) |
| High-middle SDI | 15.19(13.91–16.58) | 831(759–909) | 31.62(28.61–34.83) | 1,709(1,539–1,903) | 105.64(78.90 to 136.38) |
| High SDI | 16.60(15.93–17.25) | 971(928–1,017) | 20.97(19.98–22.46) | 1,220(1,148–1,341) | 25.71(14.73 to 37.74) |

DALY, Disability-Adjusted Life Year; SDI, sociodemographic index; CI, confidence interval.

the rate increased with age and peaked in the 95 years or older age group (26,177 per 100,000 population, 95% CI: 25,003–27,576) (S2 Table). The age structure of deaths and YLLs was similar to that of DALYs, except that the number of deaths was highest for females in the 80–84 age group and for males in the 70–74 age group (S2 Fig). Compared with 2020, females exhibited a higher percentage change than males for four indicators in 2021 (Death: 64.39% versus 57.19%; DALYs: 76.41% versus 64.13%). Furthermore, the magnitude of the percentage change was larger for children and middle-aged populations.

Regionally, the spatial distribution of incidence, deaths, DALYs, and YLLs exhibited substantial similarity but varied widely across regions (Figs 2 and S3 and S2 Table). Central Europe, Eastern Europe, and Central Asia, as well as

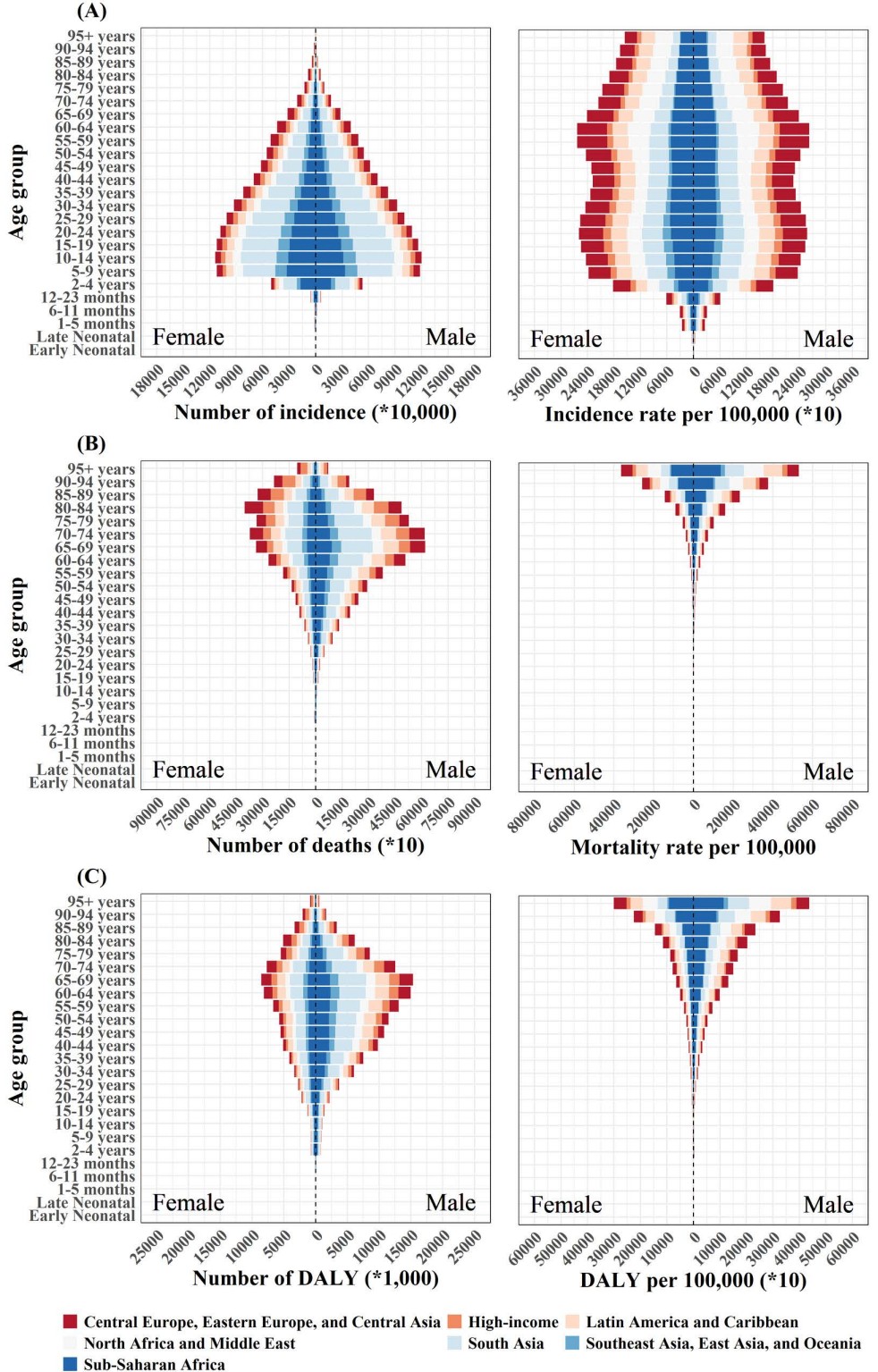

**Fig 1. Age-specific incidence (A), death (B), and DALY (C) due to COVID-19 by sex and GBD super-region, 2021.** The left column is the case number, and the right column is the rate. DALY, Disability-Adjusted Life Year.

sub-Saharan Africa had the highest ASRs due to COVID-19. However, Oceania witnessed the largest increase in ASR for all four indicators. For instance, the age-standardized incidence rate surged by 2,468.56% (95% CI: 1,140.96% to 5,216.45%) from 2020 (1,283 per 100,000 population, 95% CI: 782–2,181) to 2021 (32,964 per 100,000 population, 95% CI: 17,228–48,997). By contrast, East Asia had a reduction of 91.87% (95% CI: 89.62% to 93.64%) in incidence compared to 2020, with an ASR of only 14 (95% CI: 12–18) per 100,000 population in 2021, which was remarkably lower than that in other regions. This reduction was also observed for deaths, DALYs, and YLLs.

Fig 3 and S3 Table demonstrate health inequities in incidence, deaths, DALYs, and YLLs, respectively. Remarkable absolute and relative inequities associated with SDI were detected in the burden of COVID-19. Compared to 2020, the slope indexes for deaths, DALYs, and YLLs were significantly larger in 2021, although the difference for incidence was not statistically significant. For example, the slope index of DALYs was −2,713 (95% CI: −3,392 to −2,034) in 2020, indicating that countries with the lowest SDI had more than 2,713 additional DALYs per 100,000 population compared to those with the highest SDI, and this gap further increased to −4,044 (95% CI: −4,921 to −3,168) in 2021. Furthermore, the concentration index as a measure of relative inequity was consistently below −0.2 for all four indicators across years, suggesting an extremely imbalanced distribution of burden across countries with different SDIs. Similar patterns of inequity were also observed across the three individual components of the SDI (S4 Fig). Among sex-stratified analyses, males exhibited greater SDI-related inequity in COVID-19 burden compared to females (e.g., DALYs: −5,401 versus −2,765 for slope index; −0.282 versus −0.265 for concentration index), except for absolute inequity in incidence. In addition, the age distributions of the two health inequity indicators were not similar, with the slope index being greater in the middle-aged and older age groups, whereas the concentration index was more concentrated in the 30–79-year-olds (Figs 4 and S5). Particularly, absolute inequity for older adults surpassed that of other age groups for deaths, DALYs, and YLLs.

As expected, countries with lower SDI exhibited a disproportionate concentration of higher incidence, deaths, DALYs, and YLLs. For regions, significant slope indexes for COVID-19 were observed in Central Europe (Incidence: −20,196, 95% CI: −38,009 to −2,382; DALYs: −2,654, 95% CI: −4,865 to −443) and Latin America and the Caribbean (Incidence: −18,230, 95% CI: −27,609 to −8,851; DALYs: −2,739, 95% CI: −4,200 to −1,278), while the concentration index was generally significant in high-income North America (Incidence: −0.059, 95% CI: −0.070 to −0.048; DALYs: −0.065, 95% CI: −0.074 to −0.056) and Southeast Asia, East Asia, and Oceania (Incidence: −0.578, 95% CI: −0.760 to −0.396; DALYs: −0.598, 95% CI: −0.798 to −0.397). Compared to 2020, incidence in high-income Latin America and deaths, DALYs, and YLLs in Oceania showed substantially larger slope indexes in 2021. Conversely, relative health inequity decreased in most regions in 2021, except for the Caribbean. It is worth noting that, contrary to other regions, the high-middle SDI region unexpectedly exhibited a significant positive concentration index, indicating a heavier disease burden in more developed countries.

## Discussion

This study systematically quantified cross-country health inequities in the impact of COVID-19 by examining its incidence, death, DALYs, and YLLs across different sociodemographic strata. Notably, the findings underscore inequities in COVID-19 outcomes shaped by sex, age, and SDI. Premature death was the primary determinant of the global burden of COVID-19. Older people and males experienced higher mortality rates from COVID-19. Regionally, Central Europe, Eastern Europe, and Central Asia, as well as sub-Saharan Africa had the highest ASRs due to COVID-19. Remarkable absolute and relative inequities associated with SDI were detected in the burden of COVID-19, with low-SDI regions enduring significantly elevated incidence and DALYs rates. The absolute inequities were even further expanded in 2021 compared to 2020. Males and older individuals faced greater health inequities. These findings advance our understanding of how global inequity shapes the impact of the pandemic and underscore the importance of SDI as a predictor of health outcomes.

**(A) Incidence**

**(B) Deaths**

**(C) DALY**

**Fig 2. Country-specific incidence, death, and DALY due to COVID-19 in 2021.** The left column is the age-standardized rate, and the right column is the percentage change in age-standardized rates between 2020 and 2021. DALY, Disability-Adjusted Life Year. The basemap shapefile was from the R package "maps" version 3.4.3. https://cran.rstudio.com/web/packages/maps/index.html.

The significant health burden imposed by COVID-19 has reversed global health outcomes for the first time in the past three decades [13]. From 1990 to 2019, global health outcomes showed a steady decline in age-standardized incidence, mortality, DALYs, and YLLs rates. This can be attributed to the substantial achievements of the global health community, such as improvements in under-5 mortality and the reductions in vaccine-preventable deaths [13]. However, there was a remarkable shift in this trend from 2019 to 2021. Although only two years of pandemic data were available, the reversed trend in the global disease burden was evident from 2020 to 2021, and the AAPC of mortality showed a significant

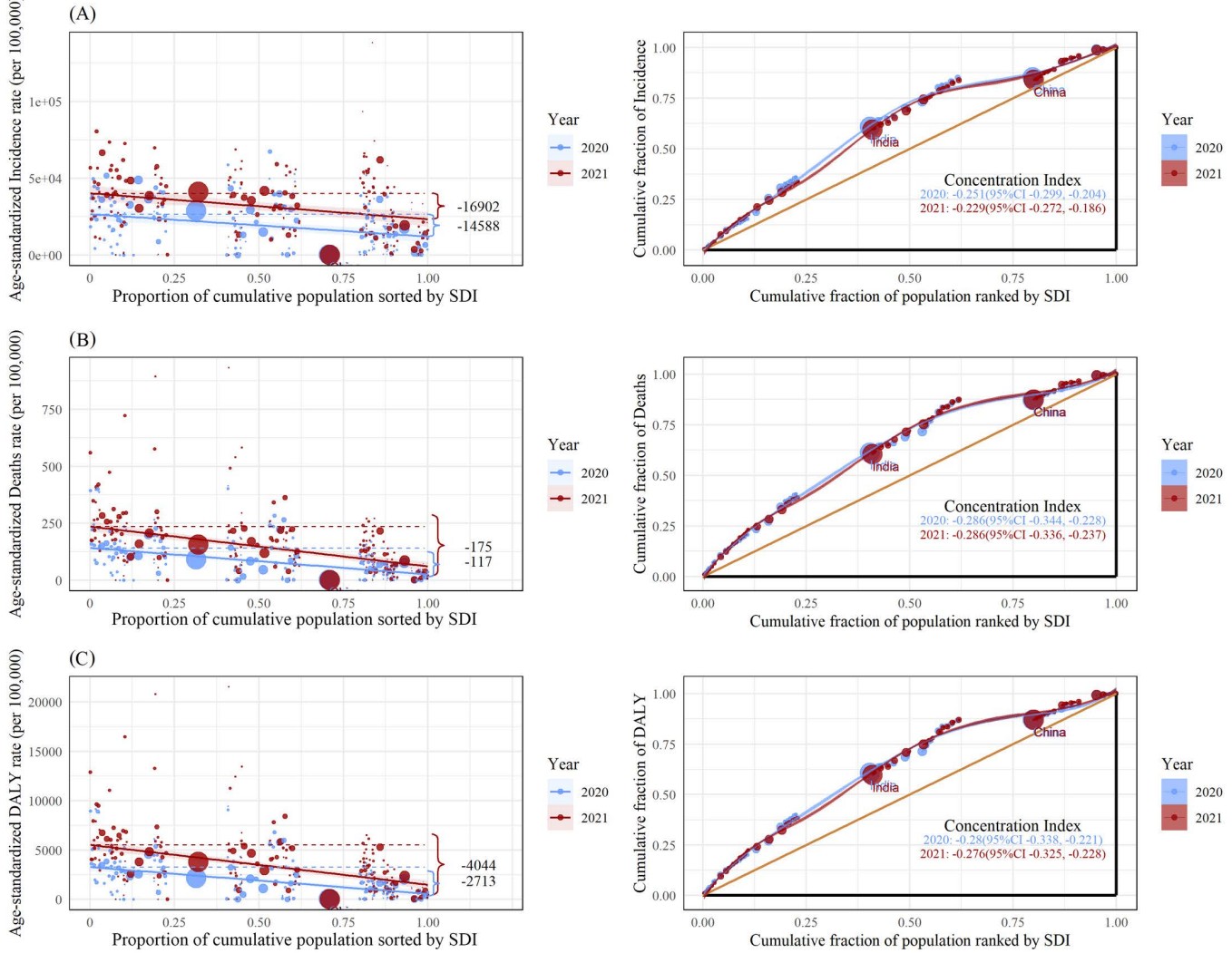

**Fig 3. Health inequity regression curves (left column) and concentration curves (right column) of COVID-19 from 2020 to 2021 across the world.** Panels **(A)–(C)** show the incidence, death, and DALY, respectively. DALY, Disability-Adjusted Life Year; SDI, sociodemographic index; CI, confidence interval.

difference compared to the pre-2019 period. Importantly, the COVID-19 pandemic served as the pivotal point for the reversal of progress, as the AAPCs during the pandemic period were not significant after excluding the cases and deaths due to COVID-19 (Table 1). In 2021, COVID-19 had become the leading cause of global DALYs, with a total of 2.28 billion incident cases and 7.89 million deaths attributed to COVID-19 [13]. These estimates differed markedly from those reported by the World Health Organization, which relied on confirmed cases reported by national surveillance systems. In contrast, our estimates were derived from the GBD 2021 framework, which integrates multiple data sources (including reported cases and deaths, hospital admissions, and representative seroprevalence surveys) to infer total infections and deaths, capturing both direct and indirect pandemic-related impacts [19]. Moreover, COVID-19 accounted for an estimated 212 million DALYs in 2021 [13]. Our analysis showed that more than 93% of this burden was attributable to YLLs, reflecting premature mortality as the dominant driver of COVID-19 impact, aligning with previous studies [22]. The possible reason is that the acute phase of COVID-19 is estimated to be relatively short, with estimates ranging from 14 days to about 28 days

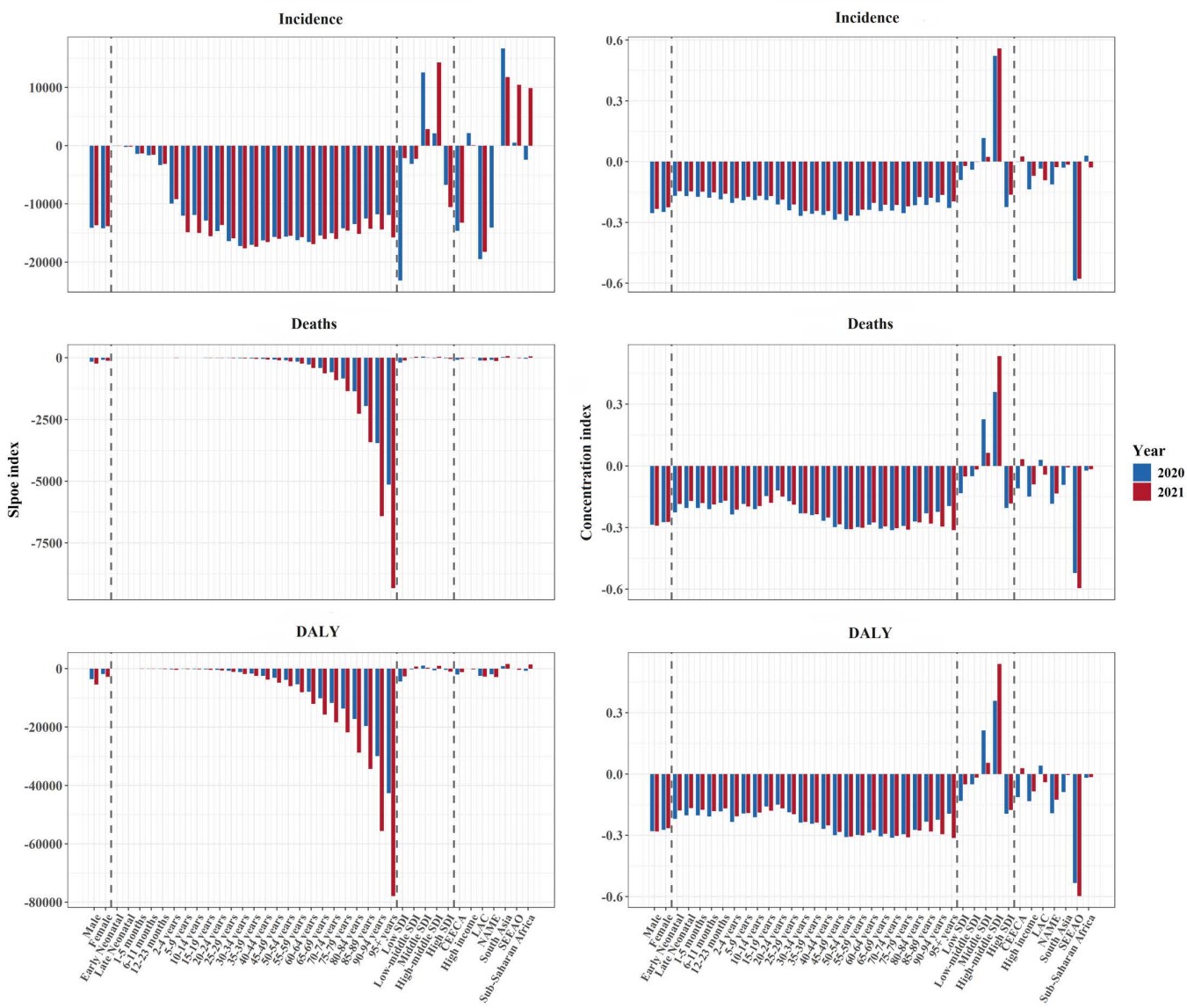

**Fig 4. Inequity indexes for the burden of COVID-19 by sex, age groups, SDI, and super-regions in 2020 and 2021.** Panels **(A)–(C)** show the incidence, death, and DALY, respectively. DALY, Disability-Adjusted Life Year; SDI, sociodemographic index.

[23]. This short period of morbidity contributes less to overall DALYs compared to the years lost due to premature death. Despite the evident impact of the COVID-19 pandemic, the global age-standardized disease burden was projected to continue declining since 2022, albeit at a slower rate than during the three decades preceding the COVID-19 pandemic [24].

While the age-standardized incidence rate of COVID-19 demonstrated a decreasing trend with advancing age among the older individuals (over 65 years), the corresponding age-standardized mortality rate increased rapidly with age. This inverse relationship could reflect a combination of heightened age-specific vulnerability and potentially limited mobility among the older people, resulting in a reduced risk of infection. Young individuals generally have more frequent social contacts than older adults, thereby increasing their exposure to SARS-CoV-2. However, age-related decline and dysregulation of immune function make older people more vulnerable to severe COVID-19 outcomes [25]. For sex-based

differences, while COVID-19 incidence was nearly equal across sexes, males experienced disproportionately higher mortality rates. This observation is consistent with previous systematic reviews indicating that males had a relative risk of 1.36 (95% CI: 1.17 to 1.59) for COVID-19 mortality compared to females [26]. The potential reasons may involve sex differences in immune responses (e.g., higher expression levels of the Toll-like receptor 7 in females) and behavioral and lifestyle factors [26,27]. Additionally, a higher prevalence of comorbidities such as hypertension and cardiovascular diseases in males likely exacerbates adverse COVID-19 outcomes [28]. These findings suggest that health interventions during the pandemic should incorporate sex- and age-sensitive approaches, particularly for reducing mortality among high-risk male populations.

A substantial regional variation in COVID-19 impact was observed, with particularly elevated age-standardized incidence and mortality rates in Central and Eastern Europe, Central Asia, and sub-Saharan Africa. These disproportionate burdens may be due to a combination of socioeconomic instability, constrained healthcare infrastructure, and low vaccination uptake. Notably, Oceania exhibited a substantial increase in the COVID-19 burden from 2020 to 2021, likely due to delayed pandemic exposure following eased travel restrictions. Conversely, East Asia achieved marked reductions in both incidence and mortality, potentially attributable to stringent public health interventions and accelerated vaccination rollouts.

Moreover, we found that regions with lower SDI experienced a more pronounced deterioration in overall health outcomes during the COVID-19 pandemic, which is consistent with previous quantitative evidence indicating significant cross-country health inequities in the burden of COVID-19 [13], which suggests that socioeconomic determinants have a pivotal influence on the uneven distribution of COVID-19's impact [6,29]. These inequities may also be attributed to other factors, such as SDI-related unequal exposure, transmission, and vulnerability [6]. For example, people in low-SDI countries are more likely to engage in non-remote work and therefore benefit less from lockdown restrictions compared to those who are able to work from home, thereby heightening their risk of exposure to COVID-19 [6,29]. Overcrowded living conditions, common in areas with high poverty rates, impede effective quarantine measures like self-isolation, which may facilitate virus transmission [30]. Additionally, populations in low SDI countries may experience a disproportionately higher prevalence of underlying disease conditions, such as respiratory diseases, obesity, or heart disease, which are identified as key risk factors for severe adverse outcomes from COVID-19 [31]. Lastly, and importantly, the limited resilience of healthcare systems in low SDI countries to manage surges in demand during the pandemic, characterized by long-term structural deficiencies in healthcare infrastructure and workforce capacity, remains a key determinant of cross-country inequities in the burden of COVID-19.

Notably, pronounced inequities in vaccination rates contribute to the significant inequities in the global burden of COVID-19. As demonstrated in S6 Fig, substantial absolute and relative inequities associated with SDI were observed in the vaccination rate of the COVID-19 vaccine. In 2021, the slope index of individuals receiving at least one dose of the COVID-19 vaccine was 56.88 (95% CI: 48.46–65.29), indicating a 56.88% higher vaccination rate in countries with the highest SDI compared to those with the lowest SDI [32]. This disparity largely reflects structural disadvantages in vaccine access. Low SDI countries generally lack sufficient manufacturing capacities for COVID-19 vaccines [33] and are therefore reliant on international vaccine supplies through market purchases or donations. However, low-income countries generally face severe financial constraints [34], and vaccine nationalism further constrains the international supplies of vaccine donations [35], as high-income countries prioritized vaccinating their domestic populations and stockpiling vaccines, rather than donating COVID-19 vaccines to curb the spread of COVID-19 in other regions, thereby exacerbating existing global health inequities [36]. Due to the absence of a coordinated global strategy, COVID-19 vaccinations in lower-income countries were delayed by several months and progressed at a slower pace, which, in turn, increases their vulnerability to COVID-19 and its variants [37]. Beyond supply-side limitations, another problem underlying vaccine inequity is vaccine hesitancy, often driven by misinformation and distrust, particularly in low-income settings, thereby compounding the problem of low vaccination uptake [38]. This hesitancy, coupled with logistical challenges in vaccine distribution, results in slower immunization progress, leaving these populations at elevated risk of severe COVID-19 outcomes.

Males experienced more pronounced inequities in COVID-19 mortality compared to females. Consistent with our results, a previous analysis indicated that males experienced disproportionately higher mortality rates from COVID-19 than females due to sex differences in immune responses and behavioral and lifestyle factors, such as a higher prevalence of smoking and alcohol consumption [26]. Systematic review evidence further confirmed that males are more likely to suffer from severe COVID-19-associated death and ICU admission [39]. Thus, the male population may require more medical resources to combat the threat of COVID-19, such as ICU equipment and ventilatory support. However, these resources are clearly insufficient in low-SDI countries, thereby exposing males to greater health inequities related to COVID-19. Additionally, socio-cultural factors play a role in sex-specific health-seeking behaviors and healthcare access, further amplifying these inequities [26,40]. Our results also revealed that absolute inequity in COVID-19 mortality increased with age. This inequity suggests that while older adults exhibit lower infection rates, they experience disproportionately high mortality risks due to the prevalence of comorbidities and potentially limited access to critical care services in resource-constrained settings [41]. These findings emphasize the urgent need for age-specific interventions to minimize preventable mortality among older populations, especially in low-SDI countries. As global life expectancy rises, reinforcing healthcare systems with health equity to meet the unique vulnerabilities of elderly populations becomes imperative for future pandemic resilience.

Through the stratification of countries based on SDI, we observed pronounced absolute inequity among low-SDI countries and significant relative inequity among high-SDI countries. This discrepancy may be attributed to differences in baseline health conditions and healthcare infrastructure. In low-SDI countries, the prominent absolute inequity largely reflects higher baseline levels of disease burden in these regions, driven by systemic deficiencies including limited healthcare access, fragile infrastructure, and insufficient public health resources. Conversely, comparable absolute differences in high-SDI countries often translate into greater relative inequities, despite generally lower overall burdens. These relative inequities are usually attributed to persistent structural barriers faced by marginalized populations, such as systemic discrimination or geographic inequities. Therefore, even in countries with well-developed healthcare systems and low disease burdens, high-SDI nations may still encounter significant inequities in health outcomes during the pandemic. Moreover, middle-to-high SDI regions presented a significantly positive inequity index, suggesting a heavier disease burden in wealthier countries within these regions. This phenomenon may reflect factors such as higher population density, increased international mobility, and varied public health responses. Conversely, low SDI regions consistently exhibited high COVID-19 mortality due to limited healthcare resources. These findings may also imply that while SDI remains a powerful determinant of pandemic vulnerability, it is insufficient on its own to fully explain cross-country inequities. Additional factors, particularly policy responsiveness and governance capacity, are equally pivotal in mitigating inequities in pandemic outcomes [42].

Understanding the impact of health inequities on COVID-19 is not only of academic interest but also holds critical public health implications. Our findings indicate that individuals living in low-SDI countries, males, and older people were more affected by COVID-19, with higher age-standardized mortality rates and greater health inequities related to COVID-19. These findings can inform equity-focused policies aimed at reducing health inequities and improving healthcare accessibility for marginalized groups, which is essential for building more resilient health systems. For example: (1) Vaccine allocation strategies should prioritize low-SDI regions and large older populations, and incorporate mobile vaccination units to enhance access in underserved areas; (2) Community-based testing and early treatment programs in urban slums and rural regions, leveraging rapid antigen tests and antiviral drugs administered by local health workers, can effectively reach marginalized populations; (3) Protective policies targeting vulnerable groups, such as paid sick leave and workplace safety regulations in high-risk occupations (e.g., transportation and sanitation), where older men are disproportionately represented, can help reduce mortality inequities associated with occupational exposure. Beyond national responses, this study also highlights the necessity of strengthened international cooperation, especially a coordinated vaccination strategy, which is crucial for the effective containment of and response to pandemics, and for addressing potential health

inequity, as the COVID-19 pandemic has demonstrated that infectious diseases have no borders between countries. Additionally, as indicated in our study, the COVID-19 pandemic has disproportionately affected marginalized populations, particularly in the resource-poor countries of subtropical and tropical regions, which also face significant burdens from neglected tropical diseases (NTDs). Our study highlights similar socioeconomic-related inequities in the burden of COVID-19 with NTDs, underscoring the necessity of enhancing healthcare systems to effectively manage and eliminate both COVID-19 pandemics and NTDs.

Despite offering valuable insights into the global impact of COVID-19, this study has several limitations. First, variations in data quality and reporting consistency across countries may have introduced potential biases. However, by synthesizing multiple data sources and employing an integrated modeling approach, GBD 2021 used predictive models incorporating multiple covariates to generate regionally consistent estimates for countries and regions with limited or missing data. These predictive models were validated and demonstrated good performance [13]. These model-based estimates cannot fully replace high-quality observational data, although they substantially minimize the risk of underestimating disease burden in underreported regions. Second, using the three-component SDI as the primary socioeconomic metric may oversimplify the complex socioeconomic contexts within different countries. Future studies could benefit from incorporating external data on alternative socioeconomic indicators to capture inequity more comprehensively. Third, future research incorporating individual-level data and multivariable models is warranted to explore potential interaction effects between key variables such as sex, age, and the SDI, which would be valuable in identifying more specific high-risk subpopulations for targeted interventions. Finally, while this study examined the acute impacts of the pandemic based on almost three decades of pre-pandemic data as well as two years of data during the pandemic, post-pandemic data were not available due to limited accessibility. Future longitudinal analyses based on accumulated data are expected to investigate the long-term impact of COVID-19 during the post-pandemic period, which will be crucial for informing public health policies.

## Conclusions

This study underscores the profound health inequities in COVID-19 outcomes across regions, age groups, and socioeconomic strata. Our findings reveal that countries with lower SDI levels bear a disproportionately higher burden of COVID-19, characterized by elevated incidence, mortality, DALYs, and YLLs. The analysis also highlights critical inequities influenced by sex and age. By elucidating the health inequities of COVID-19, this study provides a comprehensive framework for understanding how pandemics disproportionately affect marginalized populations and offers actionable insights to guide equity-focused strategies in future global health responses.

## Supporting information

**S1 Fig. The joinpoint regression analysis on age-standardized rate of COVID-19 incidence, death, DALY, and YLL.** The red points are the selected joint points. DALY, Disability-Adjusted Life Year; YLL, Years of Life Lost; AAPC, Average annual percentage change.
(DOCX)

**S2 Fig. Age-specific YLL due to COVID-19 by sex and GBD super-region in 2021.** The left column is the case number and the right column is the rate. YLL, Years of Life Lost.
(DOCX)

**S3 Fig. Country-specific YLL due to COVID-19 in 2021.** The upper panel is the age-standardized rate and the bottom panel is the percentage change in age-standardized rates between 2020 and 2021. YLL, Years of Life Lost. The basemap shapefile was from R package "maps" version 3.4.3. https://cran.rstudio.com/web/packages/maps/index.html.
(DOCX)

**S4 Fig. Inequity indexes related to SDI and its three components for the incidence, death, DALY, and YLL of COVID-19 in 2020 and 2021.** The bar graphs represent the inequity indexes and the error bars indicate the corresponding 95% confidence intervals. DALY, Disability-Adjusted Life Year; YLL, Years of Life Lost; SDI, sociodemographic index; EDU15+, mean educational attainment for those aged 15 and older; TFU25, total fertility rate under the age of 25; LDI, lag-distributed income.
(DOCX)

**S5 Fig. Inequity indexes for the YLL of COVID-19 by sex, age groups, SDI, and super-regions in 2020 and 2021.** YLL, Years of Life Lost; SDI, sociodemographic index.
(DOCX)

**S6 Fig. Inequity regression curve (A) and concentration curve (B) of COVID-19 vaccination rates across the world.** SDI, sociodemographic index; CI, confidence interval. Vaccination rate refers to the percentage of total population vaccinated with at least one dose of a COVID-19 vaccine until 31 December 2021. Data was collected from Our World in Data, available at https://ourworldindata.org/coronavirus (Accessed June 25, 2025).
(DOCX)

**S1 Table. The case number and age-standardized rates of YLLs for COVID-19 by sex and SDI in 2020 and 2021.** * Percentage change is the comparison of age-standardized rates for 2020 and 2021. YLL, Years of Life Lost; SDI, sociodemographic index; CI, confidence interval.
(DOCX)

**S2 Table. The case number and age-standardized rates of incidence, deaths, DALYs, and YLLs for COVID-19 by age and region in 2020 and 2021.** * Percentage change is the comparison of age-standardized rates for 2020 and 2021. DALY, Disability-Adjusted Life Year; YLL, Years of Life Lost; CI, confidence interval.
(DOCX)

**S3 Table. Slope and concentration indexes of the incidence, death, DALYs, and YLLs of COVID-19 by sex, SDI and region.** * P-values are comparisons of inequity indexes for 2020 and 2021. DALY, Disability-Adjusted Life Year; YLL, Years of Life Lost; SDI, sociodemographic index; CI, confidence interval.
(DOCX)

## Acknowledgments

We appreciate Gates and Melinda Foundation and the works by the Global Burden of Disease study 2021 collaborators.

## Author contributions

**Conceptualization:** Zhou Yang, Chun-Quan Ou.

**Data curation:** Xiao-Han Xu.

**Formal analysis:** Xiao-Han Xu.

**Funding acquisition:** Chun-Quan Ou.

**Investigation:** Xiao-Han Xu.

**Supervision:** Chun-Quan Ou.

**Writing – original draft:** Xiao-Han Xu, Zhou Yang.

**Writing – review & editing:** Xiao-Han Xu, Jing Sun, Jin-Lun Zhang, Zhou Yang, Chun-Quan Ou.

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
