## [Decision Letter · Decision Letter 0]

4 Jun 2025

PNTD-D-24-01908

Trends and cross-country inequalities by region, sex, age in the mortality, incidence, and disability-adjusted life years of COVID-19: analysis from the Global Burden of Disease Study 2021

Dear Dr. Ou,

Thank you for submitting your manuscript to PLOS Neglected Tropical Diseases. After careful consideration, we feel that it has merit but does not fully meet PLOS Neglected Tropical Diseases's publication criteria as it currently stands. Therefore, we invite you to submit a revised version of the manuscript that addresses the points raised during the review process.

Please submit your revised manuscript within 60 days Aug 03 2025 11:59PM. If you will need more time than this to complete your revisions, please reply to this message or contact the journal office at plosntds@plos.org. Please include the following items when submitting your revised manuscript:

We look forward to receiving your revised manuscript.

Kind regards,

Kate Zinszer

Academic Editor

Michael Holbrook

Section Editor

Shaden Kamhawi

co-Editor-in-Chief

Paul Brindley

co-Editor-in-Chief

**Additional Editor Comments :**

In addition to addressing the comments by the reviewers, please also address this following concerns:

- The justification for joinpoint analysis and including scatterplots with the breakpoints and fitted lines. It is unclear if a breakpoint at 2020 was used.

- There should be clarification on what is meant by uncertainty interval (unclear if it refers to the empirical quantiles of the distribution across countries)

**Journal Requirements:**

At this stage, the following Authors/Authors require contributions: Xiao-Han Xu, Jing Sun, Jin-Lun Zhang, Zhou Yang, and Chun-Quan Ou. Please ensure that the full contributions of each author are acknowledged in the "Add/Edit/Remove Authors" section of our submission form.

3) Some material included in your submission may be copyrighted. According to PLOSu2019s copyright policy, authors who use figures or other material (e.g., graphics, clipart, maps) from another author or copyright holder must demonstrate or obtain permission to publish this material under the Creative Commons Attribution 4.0 International (CC BY 4.0) License used by PLOS journals. Please closely review the details of PLOSu2019s copyright requirements here: PLOS Licenses and Copyright. If you need to request permissions from a copyright holder, you may use PLOS's Copyright Content Permission form.

Potential Copyright Issues:

i) Figures 2, and S2. Please (a) provide a direct link to the base layer of the map (i.e., the country or region border shape) and ensure this is also included in the figure legend; and (b) provide a link to the terms of use / license information for the base layer image or shapefile. We cannot publish proprietary or copyrighted maps (e.g. Google Maps, Mapquest) and the terms of use for your map base layer must be compatible with our CC BY 4.0 license.

**Reviewers' Comments:**

Reviewer's Responses to Questions

**Key Review Criteria Required for Acceptance?**

**Methods**

-Are the objectives of the study clearly articulated with a clear testable hypothesis stated?

-Is the study design appropriate to address the stated objectives?

-Is the population clearly described and appropriate for the hypothesis being tested?

-Is the sample size sufficient to ensure adequate power to address the hypothesis being tested?

-Were correct statistical analysis used to support conclusions?

-Are there concerns about ethical or regulatory requirements being met?

Reviewer #1: The study presents a clear objective, in a study design that is fit to address it.

No hypotheses are provided, but with this explorative aim, that is not necessary.

The study uses a well-known international data source, but apart from referring to 'updates data and improved methodologies' do not offer much explanation of how these data were gathered.

Sample size is not an issue, as nationwide data are used from around the world.

The statistical analyses are primarily simple and descriptive, but suitable for the intended objectives.

Since an existing, well-defined publicly available data source is being used, there are no concerns about ethical or regulatory requirements.

Reviewer #2: The objectives of the study are clearly articulated and the methods are appropriately used to respond to the research question. The statistical analysis is correct but the authors should justify why they used 2019 as inflection point.

Reviewer #3: The objectives are clearly articulated. The study design as articulated in the methods is not exactly the same as shown in results.

The population is well described. The size of the populations is great.

Interactions between the variables should have been included to support some conclusions. However, the analysis that was done is enough to focus the conclusions on geographic and income differences.

Methods - When did you pull the GBD 2021 data? It was updated after the initial release. Please include the date.

Page 7 - line 121 - was long term COVID-19 recognized before the end of 2021?

Page 9 lines 171 - 173 - These are widely used with GBD data but not particularly outside of it. Please provide evidence. Most studies don’t examine inequality on a linear scale (though they probably should). On line 174 is the rank the quintile?

The analysis did not include interactions between sex, age, etc. If the authors have the capability to include this in the adjusted analysis (which is not actually emphasized anywhere in the results so it isn’t clear if these results are adjusted or unadjusted for all variables), this would make the statement on page 28 lines 353-355 more accurate. Considering lines 445-452 (pages 32 and 33) specifically set up the idea that narrowing down exactly which populations need help, looking at the interactions would help policymakers know that it isn’t just males that need help but what particular subgroups need help. Is it just that males have high rates or is it something like older males need help but younger females need help? Because of the difference in the outcomes by sex, it would be very informative to discuss how the overall results (example incidence by SDI quintile) are also different by sex or why looking at that in the future would help inform policy.

**Results**

-Does the analysis presented match the analysis plan?

-Are the results clearly and completely presented?

-Are the figures (Tables, Images) of sufficient quality for clarity?

Reviewer #1: Yes, the results match the analysis plan.

The results are presented in a great level of detail (perhaps sometimes too much).

Most of the figures are good, but there are some issues like small fonts in the figure text.

Reviewer #2: The results are figures match the analysis plan.

Reviewer #3: The results are not very clearly articulated, especially the tables. The graphics make sense are are the strength of the paper. The tables are overwhelming with information and would benefit from a different form of organization or limiting to the most impactful results (significant or not).

With so much information in the tables, it is a bit difficult to follow what I’m looking at and trace back what is written to the right place in the table. This is especially true for Table 3. Is there a way to focus the tables even more on what’s the most impactful? In Table 3, what is the p-value from?

Page 10 line 196 - what do you mean “regions with different SDI levels”?

Page 12 lines 206-207 - How is premature death something that contributes to disease burden? Isn’t it the other way around?

Page 12 line 208 - Are the “four indicators” the incidence, etc? This isn’t clear. Similarly, on page 18 lines 264, is this true for the “four metrics”?

Page 18 line 265-267 - Inequity would happen by sex not within each sex (unless you specifically are looking at stratifying each sex by something such as ethnicity) so saying “inequity was greater for males than females” isn’t correct.

**Conclusions**

-Are the conclusions supported by the data presented?

-Are the limitations of analysis clearly described?

-Do the authors discuss how these data can be helpful to advance our understanding of the topic under study?

-Is public health relevance addressed?

Reviewer #1: The conclusions are well supported by the large amount of data presented.

The authors mention 3 limitations, of which I feel 2 deserve a more detailed discussion.

The usefulness of the data for understanding the topic and its public health relevance are clear in their own right, and moreover clearly discussed by the authors.

Reviewer #2: The conclusions are supported by the data presented but lacks specificity.

Reviewer #3: The methods and results do not fully match the conclusions. The discussion section emphasizes many other factors that are not included in this study without stronger connection between the two. It is very possible but needs more work.

Despite the emphasis on examining the global differences (not necessarily disparities) of the impact of COVID-19 by social, economic, and demographic groups, the method used includes very little of this information. The authors do use the SDI but this measure barely scratches the surface of either of these three categories (social, economic, or demographic). Understandably measuring some of these items and having comparable categories or items across such varied countries and regions is difficult if not impossible. However, it is very important for the authors to make this a major discussion point rather than limiting it to a single sentence (page 32 lines 437-439). To help with this, make more definitive statements in prior paragraphs in the discussion section that cover within and between country differences in socioeconomic determinants and try to better tie these paragraphs together. On the other hand, be very careful to not overstate the impact of the findings or make major discussion points that are not backed up by the evidence from the work of the authors. On lines 457-458 (pages 33) there is a “maybe” that the SDI could oversimplify the issue - IHME states on their website that it definitely does. Health issues such as hypertension and substance use were not included in the analysis, but are highlighted as very important to the outcomes in the authors’ analyses and areas for future intervention. The authors highlight smoking and drinking several times though these variables were not included in the study but are very vague about other health issues that were major contributors to poor COVID-19 outcomes (and that are also highly impacted by socioeconomic and sociocultural determinants). There are mentions of cardiovascular health but not other conditions that impact respiratory health such as COPD or asthma.

On page 27, lines 326-328, the authors state there is “compelling evidence” but do not bring back up what that evidence is.

Page 27, lines 337-338 - This sentence is not clear. What do you mean?

Page 28 lines 342-344 - Were the elderly more aware? This analysis uses data from 2020 and 2021 - is there a difference across time given the newness of the situation and the general confusion in 2020 of what to do? Nobody was sure of what to do or who was being more cautious especially across countries.

Page 28 lines 349-351 - What differences in immune responses?

On page 29, lines 378-381 - Expanding this though would really help make the emphasis on disparities stronger in this discussion.

Page 30 line 384 - What is meant by resource-limited? In this US for example, hospitals were overwhelmed whether or not they were normally resource limited. What kind of resources do you mean?

Page 30 lines 384-386 - Please quantify and better explain the relationship with non-remote work. It’s obvious to me as someone well versed in disparities and equity, but potentially not clear to other readers.

Page 30 lines 390-393 - Vaccination rates were not included in this study. This paragraph is definitely important for the discussion about why disparities exist for COVID-19 but seems out of place. Please tie it in better. For lines 394-396 - how did this exacerbate disparities? Which disparities? Line 397 - for all vaccination campaigns or just COVID-19? How does this differ from “normal”?

Is the problem “vaccine nationalism”, a lack of a coordinated strategy for vaccine distribution, or distrust of the vaccine by different populations? How does these interact?

Page 31 lines 407-408 - Is the “frequent” a raw number or a rate? Provide that here. Is there more than one manuscript to cite that points to smoking and cardiovascular disease? Why are these important for the point you’re making?

Page 31 line 409 - Remove the word “may”. This is a true statement.

Page 31 starting with line 421 - Find a better way to discuss absolute and relative differences in the discussion. This section should provide more clarity to what you mean.

Page 32 lines 441-447 - I expected the example starting on line 445 to be more specific about what this study showed in terms of the impact of these factors. Don’t be vague here.

Page 33 lines 468-470 - Burden of what?

**Editorial and Data Presentation Modifications?**

Reviewer #1: Overall the manuscript is well written and easy to read. Detailed copy-editing is particularly required in the tables/figures, where there are frequent misspellings (e.g. of abbreviations such as DALY). There are also some issues with formatting of the references.

Reviewer #2: (No Response)

Reviewer #3: There are a few sentences and statements in the introduction and discussion that seem to be the work of others and should be cited as such. Also the methods section related to GBD is mostly from the GBD website and should be cited. They also describe their set up in other papers so you could just say this is described elsewhere.

This paper should be reviewed and edited for English. There are quite a few misspellings and awkward sentences.

In the abstract, please define SDI.

Throughout the abstract, paper, graphs and tables - please spell DALYs correctly.

The confidence interval and uncertainty interval are the same thing - please be consistent in the use of one or the other. Except for early on, you mostly use the CI.

https://statmodeling.stat.columbia.edu/2022/04/05/confidence-intervals-compatability-intervals-uncertainty-intervals/

**Summary and General Comments**

Reviewer #1: This manuscript uses available data from the Global Burden of Disease Study (GBD) 21 to investigate cross-country inequalities in mortality, incidence, and DALYS of COVID-19. this builds on previous work by the GBD and extends it with a focus on inequalities. The manuscript is well written and structured, and provides a large amount of data to cover the research objective from many different angles.

I do have some issues I would like to address:

- The authors mention a total number of COVID-19 incidence cases of 2.28 billion, this seems extremely high, compared to the WHO COVID-19 dashboard which currently reports a total cumulative worldwide incidence of only 777 million. I would like to see an explanation of why these numbers are so widely different. In contrast, the total number of deaths reported here (7.89 million) is much more similar to that reported by the WHO (7.08 million).

- Throughout the article, it seems that the authors are using the terms inequalities, inequities, and disparities interchangeably. Inequity has a stronger interpretation of social justice, so I would urge the authors to study this terminology and apply it consistently.

- Both in the introduction and discussion, the authors talk about post-2019, and particularly in terms of long-term trends in health outcomes, I think this is somewhat misleading. The paper presents almost 30 years of pre-pandemic data, and compares it with two years of data gathered DURING the pandemic, and no data gathered POST-pandemic. It is very important to provide more discussion on this topic than only one paragraph in the limitations. The patterns found here represent the impact of the pandemic, but it might well be that these are short-lived and patterns will return to 'normal' in 2023 and beyond.

- The authors mention the GBD in the introduction, but do not provide much detail about what has and has not already been studied using these data. I would like to see a more thorough discussion of this, to make it clearer what the added value of the current study is, compared to previous studies using GBD 2021 data.

- Moreover, in the methods, I would like to read more about the methodology of the GBD, how do they obtain their data, and how do they try to ensure it is internationally comparable?

- The SDI is a key concept in the manuscript, so I would like to read at least who conceived it and see a reference to a more thorough description of its methodology.

- I think (some of) the data currently presented in Table 1, could be better visualized in a line graph. Now there are only 3 columns, for 1990, 2019, and 2021. Comparing these makes it difficult to conceptualize the large difference in time-span between these measurements. Using a line-graph would make visible very clearly the long-term changes, and the abrupt impact of the COVID-19 pandemic.

- In Figure 1, the colours used blend in too well to make it possible to distinguish the different regions. Either use more contrasting colours, or perhaps choose to remove the regional categories from this graph and present only overall data.

- In figure 2, it is not clear to me how useful it is to present the left column of total case numbers, in an international comparison as these are so strongly influenced by population size and characteristics. The right column of age-standardised rates is much more useful. Also, the figure legend is much too small to read.

- Table 3, overall I think it is useful and interesting to use the SDI as indicators of socioeconomic differences between countries, and using the slope index and concentration index as measures of inequality within groups. However, I do not see the added value of using both at the same time, to study the level of inequality within the same SDI category.

- In the discussion, the authors devote a long paragraph to disparities in vaccination rates. I do think this is an important issue, as possible contributor to the observed patterns, and an important source of international inequities. But this is not based on data that is presented in the current manuscript, therefore I feel it is somewhat speculative and perhaps a bit too much detail. It would be nice if the mentioned vaccination rates could be included in the manuscript in a table or figure, rather than only as mentioned in one sentence in the discussion.

- In the limitations section, the issue of variations in data quality and consistency to me is a key point for the merits of this manuscript, therefore I think it deserves a more detailed discussion. Have the GBD data been previously validated? Is there any indication of how serious these variations in quality are? The authors mention regions with underreporting, but I would like to see them interpret how this could have impacted their results and conclusions. Without underreporting, would the observed inequalities be smaller, or even larger?

- Like mentioned before, the discussion of long-term effects of the COVID-pandemic and a potential return to previously observed trends deserved a more detailed discussion.

Reviewer #2: Thanks for the opportunity to revise this manuscript. It provides a broad analysis of global COVID-19 burden and inequalities using data from the GBD study. While it uses robust methods, is well-written and responds to crucial public health questions, it suitability to PLOS NTD is questionable as COVID-19 is not neglected or tropical disease. Also, it does not affect marginalized populations as COVID-19 affects everyone. The COVID-19 pandemic, while it is an impactful topic, does not align with the scope completely. While health equity is a cross-cutting theme, the lack of direct relevance to NTD weakens its fit for PLOS NTD.

Major comments:

• The study does not address NTD. The authors should reframe the discussion to emphasize the implications for NTD affected population.

• Using data from the GBD 2021 is a strength but also a limitation. There is underreporting in low-income countries and could underestimate true inequalities. Although it is acknowledged by the author, it would be useful to discuss ho such biases might skew inequality indices. If possible, include a sensitivity analysis or discussion about this potential limitation.

• The manuscript does not disentangle the components of the SDI. Stratify SDI components in supplementary analyses or discuss their relative contributions to COVID-19 outcomes.

• Justify why 2019 was chosen as the inflection point if the pandemic began in 2020.

Minor comments:

• Is there a difference between DALY and DLAY? I understand that the whole acronym is Disability-Adjusted Life Year.

• Clarify where YLD estimates include long COVID-19 or only acute-phase disability.

• The conclusion mention “targeted interventions” but this statement lacks specificity. It would be useful to include some examples of equity-focused policies.

Reviewer #3: The subject of this paper is intriguing. The use of the GBD is great and having the data from 2020 and 2021 is great. However, this paper tries to fit too much in a small space. The strengths include the use of the GBD and the tools from the GBD (including the SDI) to compare areas that are very different. However, the discussion seems to leave the analysis behind and belongs to a different paper for the most part. I’d suggest considering splitting this paper into two and tying the ideas in the discussion closer together so it is very clear how they relate to the methods and results.

I love the idea of the work but the execution can be stronger.

As possible, please include an adjusted analysis that includes interactions and/or stratification - for example instead of just male vs female use SDI stratified by sex to make points about disparities. Creating policy based on minimal adjustment is not suggested.

Appropriately cite all work referenced from others in the introduction, methods, and discussion. Most of the methods about the GBD for example are directly from IHME.

PLOS authors have the option to publish the peer review history of their article (what does this mean?). If published, this will include your full peer review and any attached files.

Reviewer #1: No

Reviewer #2: No

Reviewer #3: No

**Figure resubmission:**
---

## [Decision Letter · Decision Letter 1]

19 Sep 2025

PNTD-D-24-01908R1
Trends and cross-country inequities by region, sex, age in the mortality, incidence, and disability-adjusted life years of COVID-19: analysis from the Global Burden of Disease Study 2021
PLOS Neglected Tropical Diseases
Dear Dr. Ou,
Thank you for submitting your manuscript to PLOS Neglected Tropical Diseases. After careful consideration, we feel that it has merit but does not fully meet PLOS Neglected Tropical Diseases's publication criteria as it currently stands. Therefore, we invite you to submit a revised version of the manuscript that addresses the points raised during the review process.
Please submit your revised manuscript within 30 days Oct 19 2025 11:59PM. If you will need more time than this to complete your revisions, please reply to this message or contact the journal office at plosntds@plos.org. Please include the following items when submitting your revised manuscript:

* A rebuttal letter that responds to each point raised by the editor and reviewer(s). You should upload this letter as a separate file labeled 'Response to Reviewers'. This file does not need to include responses to any formatting updates and technical items listed in the 'Journal Requirements' section below.
* A marked-up copy of your manuscript that highlights changes made to the original version. You should upload this as a separate file labeled 'Revised Manuscript with Track Changes'.
* An unmarked version of your revised paper without tracked changes. You should upload this as a separate file labeled 'Manuscript'.
If you would like to make changes to your financial disclosure, competing interests statement, or data availability statement, please make these updates within the submission form at the time of resubmission. Guidelines for resubmitting your figure files are available below the reviewer comments at the end of this letter.
We look forward to receiving your revised manuscript.
Kind regards,
Kate Zinszer
Academic Editor
PLOS Neglected Tropical Diseases
Michael Holbrook
Section Editor
PLOS Neglected Tropical Diseases

Shaden Kamhawi

co-Editor-in-Chief

Paul Brindley

co-Editor-in-Chief

**Additional Editor Comments:**
Reviewer #1:

- change :Page 9, Lines 169-174, the authors mention calculating a 95% confidence interval, by using the 25th and 97.5th percentiles. I assume this is just a typo and they mean the 2.5th and 97.5th

Reviewer #2:

- please do another careful review of language and grammar.

Reviewer #4:

- change : Line 55-56: Is this overall cause or infectious disease specific disease burden?

Line 126: The GBD 2021 conducted -> The GBD 2021 provided

Line 564: delete “In conclusion” since that is the title of this section :)

**Reviewers' comments:**

Reviewer's Responses to Questions

**Key Review Criteria Required for Acceptance?**

**Methods**

-Are the objectives of the study clearly articulated with a clear testable hypothesis stated?

-Is the study design appropriate to address the stated objectives?

-Is the population clearly described and appropriate for the hypothesis being tested?

-Is the sample size sufficient to ensure adequate power to address the hypothesis being tested?

-Were correct statistical analysis used to support conclusions?

-Are there concerns about ethical or regulatory requirements being met?

Reviewer #1: (No Response)

Reviewer #2: The authors have adequately addressed my concerns, and the manuscript is acceptable for publication, provided that the English is copyedited.

Reviewer #4: I am not an epidemiologist or statistician but methods appear totally appropriate to the analyses conducted.

**Results**

-Does the analysis presented match the analysis plan?

-Are the results clearly and completely presented?

-Are the figures (Tables, Images) of sufficient quality for clarity?

Reviewer #1: (No Response)

Reviewer #2: The authors have adequately addressed my concerns, and the manuscript is acceptable for publication, provided that the English is copyedited.

Reviewer #4: Very well presented based on objectives.

**Conclusions**

-Are the conclusions supported by the data presented?

-Are the limitations of analysis clearly described?

-Do the authors discuss how these data can be helpful to advance our understanding of the topic under study?

-Is public health relevance addressed?

Reviewer #1: (No Response)

Reviewer #2: The authors have adequately addressed my concerns, and the manuscript is acceptable for publication, provided that the English is copyedited.

Reviewer #4: Yes, conclusions are meaningful and appropriate based on analysis

**Editorial and Data Presentation Modifications?**

Reviewer #1: (No Response)

Reviewer #2: The authors have adequately addressed my concerns, and the manuscript is acceptable for publication, provided that the English is copyedited.

Reviewer #4: Very trivial comments:

Line 55-56: Is this overall cause or infectious disease specific disease burden?

Line 126: The GBD 2021 conducted -> The GBD 2021 provided

Line 564: delete “In conclusion” since that is the title of this section :)

**Summary and General Comments**

Reviewer #1: I commend the authors on a very thorough revision, I highly appreciate the very detailed and reference-backed responses to the issues raised by me and the other reviewer. I think the quality of the manuscript has improved substantially. I have no further issues, except that in reading the edited sections, I noticed that in the methods section, Page 9, Lines 169-174, the authors mention calculating a 95% confidence interval, by using the 25th and 97.5th percentiles. I assume this is just a typo and they mean the 2.5th and 97.5th.

Reviewer #2: The authors have adequately addressed my concerns, and the manuscript is acceptable for publication, provided that the English is copyedited.

Reviewer #4: Excellent manuscript that provides a wealth of data on global disparities in health care relative to the COVID pandemic. The manuscript is very well written and presented and the figures are superior.

PLOS authors have the option to publish the peer review history of their article (what does this mean?). If published, this will include your full peer review and any attached files.

Reviewer #1: No

Reviewer #2: No

Reviewer #4: No

**Figure resubmission:**

---

## [Editor Report · Decision Letter 2]

14 Oct 2025

Dear Dr. Ou,

We are pleased to inform you that your manuscript 'Trends and cross-country inequities by region, sex, age in the mortality, incidence, and disability-adjusted life years of COVID-19: analysis from the Global Burden of Disease Study 2021' has been provisionally accepted for publication in PLOS Neglected Tropical Diseases.

Best regards,

Kate Zinszer

Academic Editor

Michael Holbrook

Section Editor

Shaden Kamhawi

co-Editor-in-Chief

Paul Brindley

co-Editor-in-Chief

---

## [Editor Report · Acceptance letter]

Dear Dr. Ou,

We are delighted to inform you that your manuscript, "Trends and cross-country inequities by region, sex, age in the mortality, incidence, and disability-adjusted life years of COVID-19: analysis from the Global Burden of Disease Study 2021," has been formally accepted for publication in PLOS Neglected Tropical Diseases.

Best regards,

Shaden Kamhawi

co-Editor-in-Chief

Paul Brindley

co-Editor-in-Chief
